# Neutrophil Extracellular Traps and NLRP3 Inflammasome: A Disturbing Duo in Atherosclerosis, Inflammation and Atherothrombosis

**DOI:** 10.3390/vaccines11020261

**Published:** 2023-01-25

**Authors:** Puneetpal Singh, Nitin Kumar, Monica Singh, Manminder Kaur, Gurjinderpal Singh, Amit Narang, Abhinav Kanwal, Kirti Sharma, Baani Singh, Mario Di Napoli, Sarabjit Mastana

**Affiliations:** 1Division of Molecular Genetics, Department of Human Genetics, Punjabi University, Patiala 147002, India; 2Department of Neurology, MK Neuro Centre, Patiala 147002, India; 3Department of Neurology, Bhatia Hospital Neuro and Multispecialty, Patiala 147002, India; 4Department of Neurosurgery, All India Institute of Medical Sciences (AIIMS), Bathinda 151005, India; 5Department of Pharmacology, All India Institute of Medical Sciences (AIIMS), Bathinda 151005, India; 6Department of Neurological Service, Annunziata Hospital, Sulmona, 67039 L’Aquila, Italy; 7Human Genomics Laboratory, School of Sport, Exercise and Health Sciences, Loughborough University, Loughborough LE11 3TU, UK

**Keywords:** atherosclerosis, inflammation, atherothrombosis, neutrophils, monocytes, macrophages, neutrophil extracellular traps, NLRP3 inflammasome

## Abstract

Atherosclerosis is the formation of plaque within arteries due to overt assemblage of fats, cholesterol and fibrous material causing a blockage of the free flow of blood leading to ischemia. It is harshly impinging on health statistics worldwide because of being principal cause of high morbidity and mortality for several diseases including rheumatological, heart and brain disorders. Atherosclerosis is perpetuated by pro-inflammatory and exacerbated by pro-coagulatory mediators. Besides several other pathways, the formation of neutrophil extracellular traps (NETs) and the activation of the NOD-like receptor family pyrin domain containing 3 (NLRP3) inflammasome contribute significantly to the initiation and propagation of atherosclerotic plaque for its worst outcomes. The present review highlights the contribution of these two disturbing processes in atherosclerosis, inflammation and atherothrombosis in their individual as well as collaborative manner.

## 1. Introduction

Atherosclerosis is the development of plaque within the walls of arteries due to the accumulation of low density lipoproteins (LDLs), calcium, fats and cholesterol [1]. This plaque may progress to a larger size due to the involvement of immune cells causing an obstruction to the free flow of blood, which is rich in oxygen and nutrients to be delivered to different parts of the body. Other factors contributing to atherosclerosis are hypertension, tobacco smoking, diabetes, obesity and sedentary life style [2]. If the plaque remains inflamed and untreated, it causes ischemia and may rupture causing thrombus formation [3]. Generally, atherosclerosis is considered to be a problem related to heart diseases only, although it can upset any middle or large-sized artery supplying blood to any organ. Atherosclerosis is the reason for substantial morbidity and mortality for several disorders including myocardial infarction, coronary artery disease, chronic kidney disease, peripheral artery disease and stroke [1].

Thorough investigation of the signaling pathways of oxidative stress, proprotein convertase subtilisin/kexin type 9 (PCSK9), Notch signaling, Wnt signaling, mitochondrial dysfunction, pathways of cellular death, cellular excitotoxicity, dysregulated efferocytosis and many more have uncovered the pathogenesis of atherosclerosis to a large extent [1,4]. Besides these, two important cellular signaling pathways, namely the formation of neutrophil extracellular traps (NETs) and the activation of the NOD-like receptor family pyrin domain containing 3 (NLRP3) inflammasome, have started unraveling a significant contribution in the inflammatory trajectory from atherosclerosis to ischemia and to infarction and post-infarction phase.

### Atherosclerosis, Inflammation and Atherothrombosis

Atherosclerosis is the primary culprit in several diseases, and is perpetuated by inflammation to cause atherothrombosis, leading to ischemia and infarction [1,2,3]. Atherosclerosis is triggered by the accumulation and circulation of LDL particles in the blood which are rich in cholesterol, packed with phospholipids and coated with apolipoproteins [5]. These LDL enter intima from endothelium either through leaky junctions in the glycocalyx created by dying or dividing cells under the effect of transmural pressure [5] or by the process of transcytosis [6]. LDLs in intima are oxidized due to the availability of free radicals there or the catalysis of metal ions by the Fenton reaction. The endothelial layer comprises tightly placed cells that separate the blood from the vessel wall. These tight junctions may leak due to disturbed blood flow promoting the uptake of LDLs and lipoproteins. Endothelial cells are activated owing to oxidation of these LDLs and lipoproteins resulting in pronounced activation of intracellular adhesion molecule 1 (ICAM1), vascular cell adhesion molecule 1 (VCAM1) and selectins (P and E) [7]. These mediators augment the adhesion of monocytes, leukocytes and chemokine receptors such as C-C Chemokine Receptor type 2 (CCR2) and type 5 (CCR5) [8]. Chemokines help in the migration of monocyte-adhesion molecules into intima. Here, these monocytes mature into macrophages due to local macrophage colony stimulating factors (M-CSF). These macrophages exhibit scavenger receptors that bind with lipoproteins to cause foaming of LDLs (lipid laden cells). In early lesions, these macrophages are recruited; however, in the advanced form of lesions, they proliferate. These foam cells play a role in the efflux of the cholesterol or undergo apoptosis/necrosis supplementing the necrotic core with cholesterol esters. These lipid-rich macrophages incite the inflammatory cycle and provide neo-epitopes further inviting humoral and adaptive immune functions [5,6]. The dysregulated transendothelial flux of LDL causes enlargement and inflammation of intima commencing atherosclerosis [1] (Figure 1A).

The atherosclerotic plaque is propagated by the incessant accrual of lipids and oxidized LDLs. The resident smooth muscle cells (SMC) migrate from media to intima thereby thickening the plaque [1]. The inflammatory leukocytes reach the plaque site which is supplemented by extracellular components of interstitial collagen, proteoglycans, elastin and glycosaminoglycans. Components of plaque denuded from the lesion reach adjoining lymph nodes and present themselves as antigens for T and B cells [2]. T cells start localizing on plaque and Th1 cells invoke interferon gamma (IFNγ) which impairs the ability of SMCs to synthesize interstitial collagen and repair the fibrous cap over the necrotic core, further complicating the atherosclerotic plaque, whereas Th1 cells regress the lesion by producing anti-inflammatory cytokines (IL-2, IL-3 and IL-10). Macrophages and SMCs undergo necrosis and their impaired clearance (dysregulated efferocytosis) from the necrotic core makes a lipid-rich core of the atheroma [1,3] (Figure 1B). Slowly and steadily plaque is calcified by the accumulation of calcium over it, making it more vulnerable to rupture. Atherosclerotic plaques with a large lipid core and thin fibrous cap are more susceptible to rupture and incite thrombosis, whereas lesser lipid cores with a thick fibrous cap are stable [3,4]. In the case of stable plaques, another thrombotic event may emerge from lesions, called plaque erosion. This eroded lesion has been observed to have matrix components with a thin fibrous cap, less lipids and few leukocytes. It has been observed that innate immune participation through pattern recognition receptors (PRR) and polymorphonuclear leukocytes (PMLCs) amplifies the thrombotic events [1,8,9].

Generally, dysregulated extracellular matrix turnover, inflammation and coagulation along with local systemic factors progress the plaque to rupture. The components of the extracellular matrix overlay the fibrous cap over the plaque [1,10]. Tissue growth factor-beta (TGF-β) induces the synthesis of interstitial collagen. The entry of inflammatory mediators such as macrophages and T cell lymphocytes; leukocyte adhesion molecules such as ICAM-1, VCAM, selectin P and selectin E; chemokines such as monocyte chemoattractant protein-1 (MCP-1), CCR-2 and CCR-5; interleukins including interleukin-1 beta (IL-1β), interleukin-6 (IL-6), interleukin-18 (IL-18), tumor necrosis factor–alpha (TNF-α), IFNγ and cluster of differentiation 40 (CD40); cytokines such as granulocyte macrophage colony-stimulating factor (GMS-CF) and acute phase reactant; C-reactive protein (CRP) along with mediators of fibrosis such as matrix metalloproteinases (MMP), cathepsins, cystatin C and tissue inhibitor of MMPs (TIMP) render the plaque skeleton unable to sustain the pressure and the plaque ruptures [2,3,10,11]. The internal components of plaques are then exposed to the blood which invites thrombogenic molecules such as prothrombin produced by macrophages (Figure 1C). SMCs trigger thrombus formation which is the most formidable complication of atherosclerosis [6,12]. This dysfunctional endothelium and persistent thrombi generate an ischemic insult causing ischemia. These fibrin-rich thrombi activate the process of clot formation which is observed to be rich in fibrin strands, platelet clumps and NETs [12,13].

Several strategies to identify vulnerable individuals based on their immunophenotyping and surrogate end points have been suggested to alleviate the risk of atherosclerosis-driven diseases [14].Some studies have suggested that targeting epigenetic stimulators of inflammation with inhibition of Bromodomain and extraterminal motifs (BETs) can drastically reduce the expression of the worst outcomes of atherosclerosis. Studies have also suggested that therapeutic vaccination such as epitopes binding to apoB100 may reduce lesion formation. Moreover, the induction of T regulatory cells (Tregs) inhibits LDL-triggered activation of macrophages [15]. All these methods of blocking or inhibiting atherosclerotic propagation can be helpful in formulating personalized medicine for several diseases.

## 2. NETs, NETosis and Atherosclerosis

Serving as immune sentinels, neutrophils are the first to react against infectious agents and the first to reach the site of injury to catch and kill pathogens by phagocytosis to resolve inflammation (clearing pro-inflammatory stimuli) and repair tissue (promote angiogenesis) [8]. A few years ago, a new anti-pathogen strategy of neutrophils was discovered whereby on meeting a pathogen, neutrophils make a mesh-like structure called a neutrophil extracellular trap (NET) that ensnares and neutralizes pathogens [11]. NETs are web-like structures formed via decondensation of their chromatin by citrullination of arginine by peptidyl arginine deiminase 4 (PAD4) [13]. This loose chromatin becomes embedded with azurophilic granules and cytosolic proteins. The components of decondensed chromatin include predominantly positively charged proteins along with cell-free DNA and RNA. Although 70% of the proteins are histones, the rest belongs to the cytoplasm, metabolic pathways and cytoskeleton. Almost 20 proteins have been identified in the NET proteome (NETome) that participate in NET formation. These include neutrophil elastase (NE), proteinase-3 (PR3), myeloperoxidase (MPO), Cathepsin G, Keratinocyte transglutaminase, factor XIIIa, alpha-defensins and citrullinated histones (ctH) [16].

When neutrophils fail to resolve inflammation by phagocytosis and pro-inflammatory stimuli are non-subsiding and incessant then NETs are formed and thrown on the microbes (pro-inflammatory stimulus) either by breaking the plasma membrane with the pore-forming protein Gasdermin D (GSDMD) causing the death of the neutrophil (suicidal NETosis) or by transporting these NETs by membrane blebbing or vesicular exocytosis (vital NETosis) [10,13]. NETosis is the process by which the neutrophil expels its nuclear material outside the cell; however this term was earlier used for neutrophil death (Figure 2).

NET formation or NETosis is initiated by several triggers; otherwise, resting neutrophils are non-inflammatory and do not undergo NETosis [8]. Vital NETosis is observed mostly during infection rather than sterile injury, whereas suicidal NETosis is associated with sterile and noninfectious complications [13]. Several stimuli have been observed to initiate the formation of NETs such as phorbol-12-myristate-13-acetate (PMA) [17], bacterial toxin; ionomycin, lipopolysaccharides (LPS) [10], some cytokines such as IL-1β, TNFα and IL-8 [11], microbe size [18], activated platelets [19], reactive oxygen species (ROS) burst [20], histone acetylation [21], etc.

Atherosclerosis is considered to be the chief culprit in the pathology of several complex disorders, and is propelled by vascular inflammation [1]. The inflammatory trigger by lipid-rich foam cells in atherosclerosis is considered to be the central event when these accumulate in the subendothelial area of an injured artery [1,3,5]. In order to clarify whether NETs are formed during and contribute to atherosclerosis, earlier studies have shown that neutrophils were either present with condensed nuclei or were luminar rather than lesional, suggesting that neutrophils are less likely to participate in atherosclerosis development [22,23]. In essential hypertension patients, abundant NET formation was observed but when they were treated with angiotensin II (AngII), NETs were substantially reduced [24]. It has been observed that a mouse knockout for *ApoE-/-* expresses heightened NET formation and interferon-alpha (IFN-α) expression in atherosclerotic arteries. When these mice were injected daily with Cl-amidine, which is an inhibitor of the PAD4 enzyme, recruitment of neutrophils and macrophages into intima was significantly reduced, hence mitigating NET formation and reducing atherosclerotic load by delaying carotid thrombosis [22]. This suggests that PAD4 is a paramount enzyme for histone citrullination and recruitment of NETs during atherosclerosis. Another study demonstrated that NETs are not formed in mice with NE blocked in *Klebsiella pneumoniae* infection suggesting NE is vital for NET formation [20]. They observed that during an ROS burst, NE sheds off from azurophilic granules and moves to the nucleus for chromatin decondensation. It has also been observed that MPO significantly induces NET formation in *Candida albicans* infection as neutrophils from MPO-deficient patients fail to form NETs [25]. Most of these studies have been carried out on murine models, and thus PAD4-driven NETosis has been shown to occur in experimental mouse models only; however, in humans it does not influence fatty streak formation or increasing plaque size [26]. However, the same study also observed that it participates in the atherothrombotic advancement of intimal lesions prone to plaque erosion. Plaque erosion is a complication where flowing blood within arteries does not disrupt the cap of the atherosclerotic plaque but an acute thrombus is eroded from intima, where endothelial cells are damaged (endothelial denudation). Another study examined NET formation and its contribution to atherogenesis using a myeloid-specific deletion of PAD4 in *ApoE-/-* knockout mice [27]. The authors proposed strongly that NETs promote atherosclerosis in both murines and humans and this NET-driven atherogenesis is governed by PAD4, because in their experiment of PAD4 deletion in myeloid cells, a reduced NET formation and attenuated inflammatory response were observed [27].

NETs and NETosis not only participate in atherosclerosis but also contribute to thrombus formation. NETs induce a scaffold of DNA that exhibits a red blood cell (RBC)-rich thrombus along with von Willebrand factor (vWF), fibronectin and fibrinogen in experimental deep venous thrombus in baboons [12]. This inference is corroborated by a study in humans showing that activated platelets interact with neutrophils to generate tissue factors that provoke neutrophils to induce thrombogenic signals promoting atherogenesis in ST-segment elevation acute myocardial infarction (STEMI) [28]. NETs participate significantly in prothrombotic signaling by triggering the oxidation of LDLs, generating ROS, endothelial dysfunction, apoptosis, fibrin-formation-induced platelet aggregation, accumulation of vWF and fibrinogen [29]. NETs are observed to interact with inflammatory platelets to promote thrombosis via immune-related GTPase family M protein (IRGM) and its orthologs [30]. Carriers of the homozygote TT genotype of the R262W polymorphism within the Src homology 2B (SH2B) protein 3 (LNK/SH2B3) gene show augmented platelet–neutrophil aggregation leading to heightened atherosclerosis and atherothrombosis in an oxidized phospholipid (oxPL)-dependent manner [31]. Alluding to contradictions and confusions, a remarkable piece of research has answered three important queries related to the role and relevance of NETs and NETosis in an experimental murine model of atherosclerosis [32]. First, they incubated neutrophils with cholesterol crystals and observed that cholesterol crystals prompt neutrophils to synthesize NETs and undergo suicidal NETosis. Second, to investigate whether NETosis participates in atherosclerosis, NETs were observed abundantly within atherosclerotic lesions of aortic roots in mice lacking ApoE (*ApoE-/-)* who were nurtured with high-fat diets (HFD) for 8 continuous weeks. ApoE is the master player of reverse cholesterol transport, whereby it carries and transfers a larger volume of LDLs to the liver, and then these LDLs are transported to bile [33]. Therefore, its absence (*ApoE-/-*) caused hypercholesterolemia which led to an accumulation of leukocytes and plaque formation. This suggests that during hypercholesterolemia, neutrophils make NETs abundantly contiguous to cholesterol crystals. Third, to understand whether NETosis contributes to atherogenesis, they developed triple-knockout mice for ApoE and two important components of the nuclear material expelled by NETs, i.e., NE and PR3 (*ApoE^-/-^/NE^-/-^/PR3^-/-^*), and compared these with *ApoE-/-* mice. It was observed that the aortic roots of triple-knockout mice had no NET formation, reduced levels of IL-1β and fewer lesional T cells which produce cytokine IL-17. IL-17 is observed to perpetuate inflammation by inviting other pro-inflammatory cytokines such as TNF-α, IL-1β and IFN-γ [34]. This demonstrates that components of NETs, namely, NE and PR3, are required for inflammation in atherosclerosis that propels atherogenesis. To corroborate this finding, they further injected *ApoE-/-* mice with DNase, an enzyme that neutralizes DNA material, and observed that the lesion size was significantly reduced (by approximately three times), whereas lesion size was unaffected after injecting DNase into *ApoE^-/-^/NE^-/-^/PR3^-/-^* mice, who had no NET formation. It suggested that even during hypercholesterolemia caused by *ApoE^-/-^* absence, no NETs were formed because NE and PR3 were neutralized by the DNase, proving that these two are paramount for NET formation. Furthermore, this study clarified that cholesterol crystals invoke neutrophils to form NETs and exercise NETosis, components of which prime macrophages to initiate NLRP3 inflammasome activation, finally converting immature forms of IL-1β (pro-IL-1β) and IL-18 (Pro-IL-18) to mature forms and release them with the help of pore-forming Gasdermin D into the extracellular space [13].

## 3. NLRP3 Inflammasome Activation and Atherosclerosis

NLRP3 is present in the cytoplasm as an inactive protein but is activated on sensing danger from several cellular triggers [9,35]. NLRP3 inflammasome activation has been observed to play a central role in initiating the inflammatory cascade in several diseases [36]. It is a multiprotein complex that contains an adapter (ASC or PYCARD), a receptor (NLRP3) and an effector (pro-caspase-1) along with domains such as telomerase-associated protein 1 (TP1 or NACHT), neuronal apoptosis inhibitory protein (NAIP), N-terminal pyrin domain (PYD) and leucine-rich repeat (LRR). On sensing damage or danger signals by NACHT, it triggers signaling where ASC forms speck-like clusters and pro-caspase 1 (Pro-CASP1) is recruited to the ASC speck clusters. ASC and Pro-CASP1 cleave proteolytically the active caspase-1 (CASP-1), which matures pro-IL1β to IL-1β and pro-IL18 to IL-18. During this maturation, CASP-1 induces pyroptosis, which is a form of lytic cell death triggered by the formation of plasma membrane pores by gasdermin D, leading to a flux of ions (K^+^ and Ca^2+^) and releasing mature IL-1β and IL-18 into the extracellular space [37] (Figure 3).

It is well known now that for the activation of the NLRP3 inflammasome, two molecular signals are required. First, a nuclear factor kappa B (NF-κB)-dependent priming signal promotes the upregulation of IL-1β and NLRP3, and then a second signal triggers the oligomerization or activation of the NLRP3 inflammasome [38]. During atherosclerosis, cholesterol crystals evoke NETs to provide the first priming signal for macrophages, whereas the production of pro-IL-1β provides the second signal for the activation of the NLRP3 inflammasome and the release of mature cytokines. The NLRP3 inflammasome is triggered by different danger signals such as cholesterol crystals in atherosclerosis [39], uric acid crystals in gout [40] and amyloid-beta in Alzheimer’s [41]. It is also activated in response to other triggers such as a reduced K^+^ concentration in the cytoplasm resulting in P2X purinoceptor 4 (P2X4) receptor-mediated K^+^ efflux [42]. Necrotic cells of the ischemic core discharge Ca^2+^ which increases in extracellular spaces prompting an increased influx and decreased efflux of Ca^2+^ [43]. Extensive Ca^2+^ influx induces cytochrome C dislocation which impairs the mitochondrial function of ATP production leading to ROS generation [44], which triggers NLRP3 inflammasome activation [45].

A pioneer work by Duewell et al. [39] revealed that cholesterol crystals are taken up by macrophages, whereby they are degraded in phagosomes and transferred to lysosomes. These undegraded cholesterol crystals (undegraded because of their size and chemistry) invoke rupture of the phagolysosomal membrane releasing lysosomal cysteine protease cathepsin B, which is taken up as a danger signal for the priming and activation of the NLRP3 inflammasome and release of mature IL-1β. Another study showed that complement component 5a (CC5a) along with TNF-α invokes a signal for cholesterol-crystal-induced NLRP3 inflammasome activation [46]. Undegraded cholesterol crystals released after phagolysosomal membrane breach trigger the priming signal for the NLRP3 inflammasome to produce a premature form of IL-1β (pro-IL-1β) which is considered to be the activation signal for the oligomerization of the NLRP3 inflammasome to release mature IL-1β [32]. Basic calcium phosphate crystals (BCPC) are considered to be present with inflammatory macrophages in developing atherosclerotic plaque, and are internalized by macrophages [47] and initiate signals for the activation of the NLRP3 inflammasome [48]. Fatty acid palmitate promotes neointima formation by upregulation of inflammatory pathways and exerts a pro-inflammatory effect on vascular smooth muscle cells by stimulating the expression of C-reactive protein (CRP), TNF-α and inducible nitric oxide (iNOS) [49]. This palmitate prompts NLRP3 inflammasome activation through mitochondrial ROS production and lysosomal degradation [50]. Expounding on several triggers of NLRP3 inflammasome activation, two things are equivocally believed. First, NLRP3 is required as a general sensing receptor for all sorts of cellular debris. Second, many stimuli (K^+^ efflux, Ca^2+^ influx, cathepsin B and ROS formation) are generated due to the breakdown of cellular organelles.

Almost 20 years ago, the NLRP1 inflammasome was discovered, which was considered to be present in leukocytes and activated by PYCARD to release mature caspase1 and 5 [51]. Later several inflammasomes such as the NLR Family CARD Domain Containing 4 (NLRC4), Interferon-inducible protein 2 (AIM2), Pyrin and Interferon inducible protein 16 (IFI16) were discovered [52]. Earlier, when research on NOD-like receptors (NLRs) was in its infancy, it was believed that NOD-like receptor proteins (NLRPs) create inflammasomes only through caspase-1 activation, which was termed the “canonical inflammasome”. Then later, it was observed that inflammasomes can be formed by NLRPs through other methods such as with alternate caspases (caspase-11), which sense intracellular lipopolysaccharide (LPS) as a danger signal [53]. This was termed the “non-canonical inflammasome”. NLRP3 can trigger both types of inflammasome activation.

The role of NLRP3 inflammasome activation in atherosclerosis and atherogenesis can be understood by the role of two terrible cytokines produced by it, i.e., IL-1β and IL-18. Inflammation itself does not initiate atherogenesis; rather it invites several intermediaries, especially cytokines, which mediate both inflammation and immune signaling. IL-1β plays both of these roles as a key messenger for propelling inflammatory signaling to atherothrombosis [54].

A study investigated the role of IL-1β in the progression of atherosclerosis by engineering double-knockout mice (*ApoE^-/-^/IL-1β^-/-^)* [55]. The study revealed that the area and size of the lesion at the aortic sinus decreased by 30% in *ApoE^-/-^/IL-1β^-/-^* mice of 12 to 24 weeks of age when compared with *ApoE^-/-^/IL-1β^+/+^* mice. Similarly, a significant reduction in VCAM-1 and MCP-1 mRNA levels was evident in *ApoE^-/-^/IL-1β^-/-^* mice. The results suggested that *IL-1β* plays a significant role in the progression of atherosclerosis by downregulating VCAM-1 and MCP-1 expression in an injured aorta. Lately, it has been revealed that IL-1β helps in leukocyte accumulation and recruitment into atherosclerotic aortas, suggesting its role in speeding up atherosclerosis [56]. Moreover, IL-1β expression drives vascular calcification, angiotensin II (AngII)-induced hypertension and vascular remodeling [57]. Therefore, strategies for blocking IL-1β through inhibitors and agonists have proved to be very beneficial for atherosclerosis-driven diseases [58]. Similarly, IL-18, another proinflammatory cytokine produced by the activation of the NLRP3 inflammasome, is expressed in macrophages and plays a significant role in inflammatory and immune signaling by the synthesis of T cells, natural killer cells and IFNγ [59]. A study has revealed that it is rampantly present in atherosclerotic plaques and influences plaque destabilization [60]. Another study comprising knockout mice for *ApoE^-/-^* and *IL18^-/-^* revealed that this double knockout had reduced lesion size and lesser lesion composition [61]. It suggests that even in hypercholesterolemia, the absence of IL-18 attenuates lesion size and its concomitants. Another study explained its role in plaque destabilization by setting up an inflammatory hyper-response after binding to interleukin-18 receptor alpha chain (IL-18Ra) through NF-kB [62]. The CANTOS (Canakinumab Anti-inflammatory Thrombosis Outcome Study) research has substantiated the clinical relevance of blocking IL-1β and IL-18 for curtailing atherothrombosis in post-acute myocardial ischemia [63]. Nevertheless, a proposition that in advanced lesions, especially in the fibrous cap, IL-1β is atheroprotective for outward vessel remodeling and stabilizing the plaque by maintaining the thickness of fibrous cap and collagen content, cannot be ignored [64].

## 4. NETosis-NLRP3 Inflammasome Activation Link: A Maleficent Crosstalk

Clinical research on the signaling pathways leading to atherosclerosis has suggested that components of NETosis evoke NLRP3 inflammasome activation, which releases the proinflammatory cytokines IL-1β and IL-18. Both of these cytokines are very harmful for stimulating the atherosclerotic plaque to atherothrombosis and its rupture leading to ischemia. A study has revealed that monocyte-derived macrophages generate strong signals for inflammasome activation when they are incubated with NETs and then with cholesterol crystals [32]. This shows that NETs prime macrophages to produce pro-IL-1β and cholesterol crystals induce phagolysosomal damage due to internalization by binding to CD36 (a glycoprotein on the plasma membrane of macrophages). When macrophages ingest cholesterol crystals as cellular debris, they destroy them into phagosomes; these phagosomes deliver the degraded contents to lysosomes (phagolysosomes) where they are further degraded by acid hydrolases. These hydrolyzed contents of cholesterol crystals may cause the rupture of the phagolysosome membrane and the release of its contents into the cytoplasm which otherwise are exported outside the cell by membrane transporters [39,65]. These undegraded cholesterol crystals released into the cytoplasm due to phagolysosomal membrane rupture, are taken as danger signals which are sensed by PRRs, termed danger-associated molecular patterns (DAMPs) and pathogen-associated molecular patterns (PAMPs). NLRP3 is a significant member of the innate immune system which is a PRR and is activated in response to these danger signals. During NLRP3 inflammasome activation, it transposes ASC and regulates caspase-1 (CASP1) stimulation, which further proteolytically cleaves the mature forms of the proinflammatory cytokines IL-1β from pro-IL-1β and IL-18 from pro-IL-18. Present abundantly in atheromatous lesions, both IL-1β and IL-18 help in plaque development and its progression to ischemic stroke [66]. The consequent release of undegraded cholesterol crystals into the cytoplasm sets the wheel of inflammatory cascade (activating NLRP3 inflammasome) in motion through cycles of internalization of cholesterol crystals and continued phagolysosome rupturing (Figure 4). During acute inflammation, resolvins and selectins induce neutrophil apoptosis; they are engulfed and phagocytosed by macrophages in an effort to resolve inflammation and improve healing [67]. Hence, it is worthwhile to infer from the experimental evidence that cholesterol crystals incite both neutrophils and macrophages to initiate pronounced inflammation (NETosis and NLRP3 inflammasome activation) and destroy the neutrophil–macrophage amity which is otherwise anti-inflammatory, pro-healing and pro-repair [67].

To understand whether the NLRP3 inflammasome promotes NET formation during atherosclerosis, mice deficient with Abca1/g1 in myeloid cells were engineered with Nlrp3/caspase1/caspase11 deletion and bone marrow cells of these mice were transported to *Ldlr^-/-^* mice, which were fed with Western-type diet (WTD) [68]. They observed that lesion size significantly decreased in myeloid Abca1/g1-deficient *Ldlr^-/-^* mice due to Nlrp3/caspase1/caspase11 deficiency. Abca1/g1-deficient myeloid cells enhanced the cleavage of caspase 1 in splenic monocytes and macrophages and also in neutrophils to induce NET formation in atherosclerotic plaques. This indicates a cycle of heightened inflammation whereby the amassed cholesterol induces NLRP3 inflammasome activation which further promotes NET formation and NETosis in atherosclerotic plaques [68]. Another study has substantiated this inference and demonstrated that the canonical NLRP3 inflammasome plays a significant role in NETosis in sterile conditions [69]. This study has three important findings: first, PAD4, which is required for histone citrullination and decondensation of chromatin during NETosis, is required for NLRP3 oligomerization by mediating protein levels of ASC and NLRP3 post-transcriptionally. Second, genetic ablation of NLRP3 results in impaired signaling; hence, it significantly reduces NET formation in mouse neutrophils suggesting that NLRP3 components help in nuclear and plasma membrane rupture. Third, NLRP3 inhibition with a pharmacological antagonist attenuates NETosis in both mouse and human neutrophils. This study has shown that NLRP3 insufficiency leads to lesser NET formation in the thrombi of an experimental mouse model of deep venous thrombosis.

All of this suggests that PAD4 is required for NLRP3 inflammasome activation which propels NETosis both in vivo and in vitro under non-sterile conditions. Interestingly, IL-1β, a final product of NLRP3 inflammasome activation, is observed to play a significant role in NET formation and induces NETosis in systemic inflammatory response syndrome (SIRS) and abdominal aortic aneurysms (AAA) [11,70]. Moreover, cytokine IL-18 enhances the influx of Ca^2+^ into neutrophils which generates mitochondrial ROS and induces NET formation [71]. It has been demonstrated that similar processes are evident in NLRP3 inflammasome activation and in the generation of NETs [72]. Inflammasome activation proceeds to osmotic swelling of the cells, cell necrosis and finally the release of proinflammatory IL-1β, IL-18 and some DAMPs such as interleukin 1 alpha (IL-1α), high mobility group box 1 (HMGB1) proteins and ATPs. During this process, activated CASP1/11 cleaves GSDMD which neutralizes the membrane via pore formation and eventually cause pyroptosis, an inflammation-stimulated type of cell death. A similar process is evident in neutrophils where the NLRP3 inflammasome activation triggers the cleavage of GSDMD which helps in the degradation of the granular membrane, decondensation of chromatin, dissolution of the plasma membrane and expulsion of the components of NETs [72].

On the other hand, NET components including cfDNA stimulate NLRP3 inflammasome in macrophages [73]. Three significant inferences are evident from the analysis of these previously mentioned research reports. First, PAD4 is the hallmark for NET generation, and it also stimulates NLRP3 inflammasome activation. Second, cholesterol crystals trigger priming and activating signals for NLRP3 inflammasome activation and also initiate NET formation and prompt NETosis. Third, components of NETs (dead mitochondria and cfDNA) prime NLRP3 to activate the inflammasome, whereas IL-1β and IL-18 stimulate neutrophils for NET formation. Although it was known that NLRP3 inflammasome activation in macrophages is responsible for delayed wound healing in diabetic mice, it has been revealed that NET production in diabetic wounds triggers NLRP3 inflammasome activation and releases IL-1β in macrophages causing a sustained inflammatory response and prolonging wound healing [74]. Furthermore, NETs elicit signals for ROS generation triggering thioredoxin interacting protein (TXNIP) activation and both of these induce NLRP3 inflammasome activation. On summing up, one may reason to believe that both NLRP3 inflammasome activation and NETosis are triggered simultaneously, primarily by cholesterol crystals, and together these processes not only participate in triggering atherosclerosis but also perpetuate it with inflammation and exacerbate it with atherogenesis, finally leading to ischemia and infarction.

## 5. Future Directions

It is beyond doubt that our understanding of mediators that propel inflammation on atherosclerosis has improved considerably over last few years but translating this knowledge from bench to bedside still needs to be enhanced. The complex etiology coupled with several phases of atherosclerotic lesion and more so their different responses to inflammation-resolving drug targets have complicated its utility for improving anti-atherosclerosis strategies. Future studies based on multiomics platforms should weigh the collaborative contribution of these inflammatory perpetrators against trained immunity (de facto innate immune memory) and should develop some quantifiable measure stratified with risk severity so that their genetic or pharmacological inhibition may prove beneficial for specific stages of atherosclerotic lesions showing recuperative dose response. Gene expression studies along with clinical data sets in atherosclerosis may jointly be used to develop polygenic risk scores which can offer risk-stratified quantitative measures for specific immunophenotypes. Such strategies for harnessing gene-immune signatures may be the future of personalized medicine, whereby a formidable impact of the NETs–NLRP3 inflammasome nexus on atherosclerosis and its affiliated worst outcomes can be curtailed.

## Figures and Tables

**Figure 1 vaccines-11-00261-f001:**
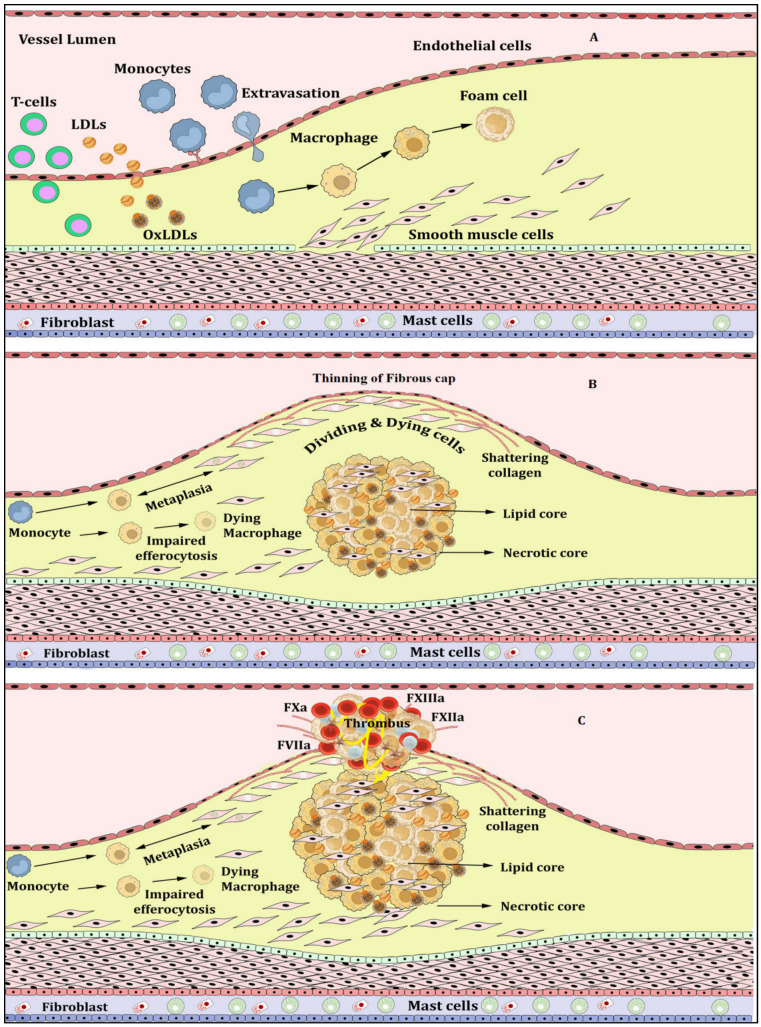
Initiation of atherosclerosis (**A**), progression (**B**) and rupture of atherosclerotic plaque (**C**).

**Figure 2 vaccines-11-00261-f002:**
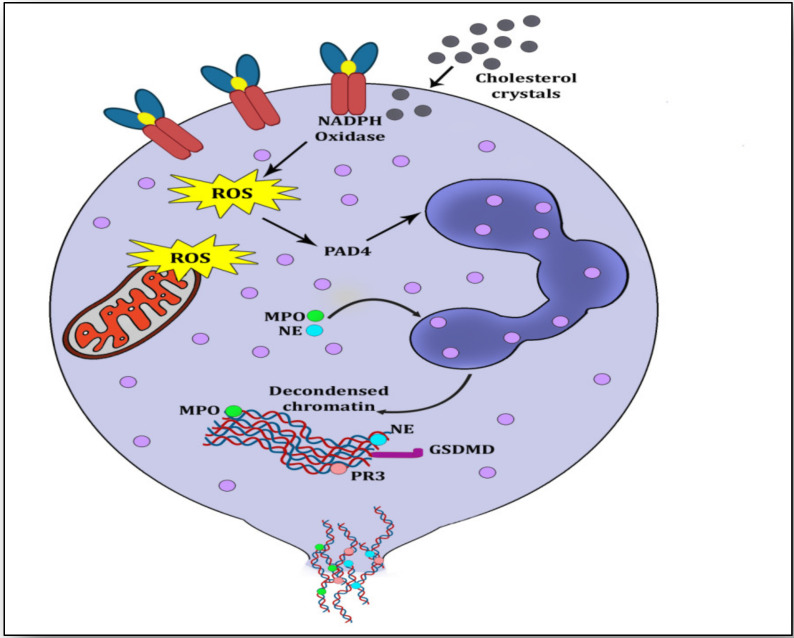
Overview of suicidal NETosis. Cholesterol crystals interact with free radicals and generate NADPH-oxidase-induced reactive oxygen species (ROS). ROS stimulate peptidyl arginine deiminase 4 (PADI4) to citrullinate arginine resulting in loosening of chromatin from histone. Myeloperoxidase (MPO) and neutrophil elastase (NE) migrate to the nuclear membrane for its rupturing by further decondensation of the chromatin. This decondensed chromatin exhibits a mesh-like structure called neutrophil extracellular trap (NET), which is ejected into the cytoplasm, where it is embedded with azurophilic granules and cytosolic proteins. Finally, this NET is ejected through the membrane rupturing of the neutrophil and causing its death.

**Figure 3 vaccines-11-00261-f003:**
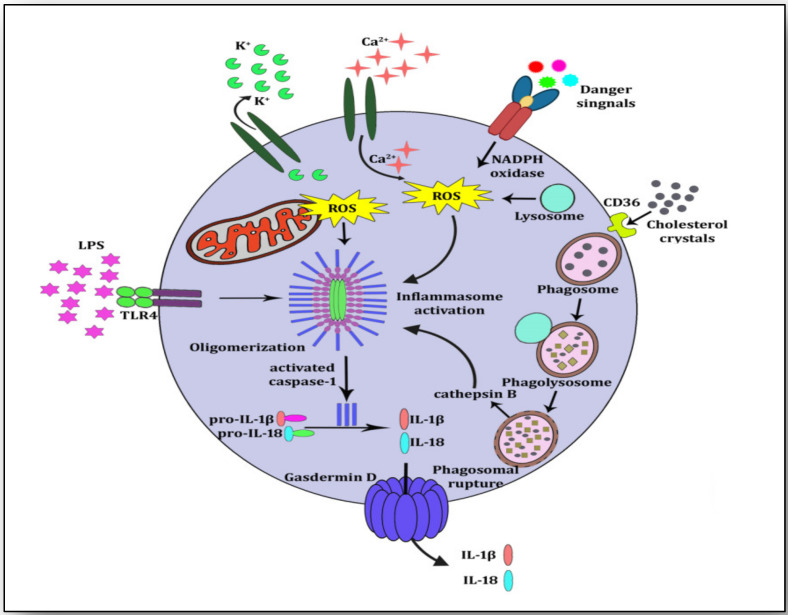
NLRP3 inflammasome activation. Cholesterol crystals are internalized by CD36 and taken by phagosomes for phagocytosis. Cholesterol crystals are broken down and lysosomes attach to phagosomes to form phagolysosomes. Because of the size and chemistry of the cholesterol crystals, the phagolysosome ruptures and undegraded crystals along with cathepsin B are exposed in cytoplasm. This is the priming signal for NLRP3 inflammasome activation. Other signals such as K^+^ efflux, Ca^2+^ influx and lipopolysaccharide (LPS) may also trigger the NLRP3 inflammasome. Immature forms of IL-1β and IL-18 (pro-IL-1β and pro-Il-18) are proteolytically cleaved by activated caspase-1. These mature cytokines IL-1β and IL-18 are released into the extra cellular space by pore-forming Gasdermin D.

**Figure 4 vaccines-11-00261-f004:**
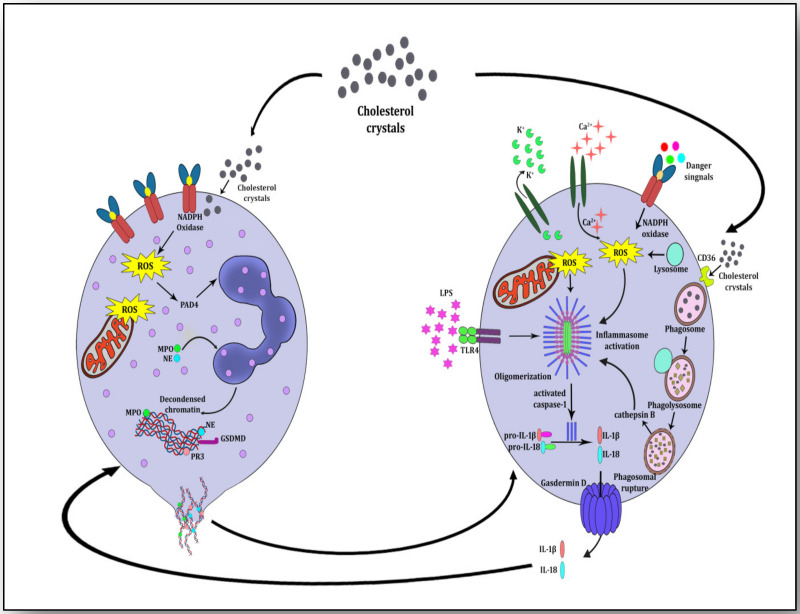
Neutrophil-macrophage crosstalk for mutually inciting atherosclerosis and igniting inflammation. Components of NETs (dead mitochondria and cfDNA) prime macrophages and components of NLRP3 inflammasome activation (IL-1β and IL-18) induce neutrophils to form NETs. Cholesterol crystals evoke signals for both neutrophils and macrophages for the development of NETs and prime NLRP3 inflammasome by interacting with reactive oxygen species (ROS) and CD36-mediated internalization, respectively.

## Data Availability

Data collected from different research papers for writing this review paper are available from the corresponding author, and can be shared on request.

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
