# Peer review of "Neutrophil Extracellular Traps and NLRP3 Inflammasome: A Disturbing Duo in Atherosclerosis, Inflammation and Atherothrombosis"

_vaccines, 2023, doi:10.3390/vaccines11020261_

Round 1

Reviewer 1 Report

Singh et al. provide concise review on the importance and clinically relevant NET and NLRP3 driven Atherosclerosis and Atherothrombosis. The review is well written with the content and figures are good and explanatory to the readers. However, I strongly feel few very important references are missing like: PMID: 35504280 , PMID: 28209798, PMID: 29045897, PMID: 35102320, PMID: 31280097, PMID: 36580725, PMID: 34846914, PMID: 34324440.

There are more relevant articles or review to the related topic and could be word limit or other constraint though inclusion will expand the knowledge of current review.

Line 311- exposed is a right word here?

Author Response

We are very grateful to the reviewers for pointing out very important and informative issues for the betterment of this manuscript. Please see point wise response in bold face.

Reviewer 1

Singh et al. provide concise review on the importance and clinically relevant NET and NLRP3 driven Atherosclerosis and Atherothrombosis. The review is well written with the content and figures are good and explanatory to the readers. However, I strongly feel few very important references are missing like: PMID: 35504280, PMID: 28209798, PMID: 29045897, PMID:35102320, PMID: 31280097, PMID: 36580725, PMID: 34846914, PMID: 34324440.

We have included all these 8 important papers in the text.

There are more relevant articles or review to the related topic and could be word limit or other constraint though inclusion will expand the knowledge of current review.

Yes, we agree on this point. By including more information on pharmacological inhibitors, drug targets and genetic ablations to block the NLRP3 inflammasome and NETs generation, it will consume 4-5 pages and many references.

Line 311- exposed is a right word here?

Sir this word ‘exposed’ has been replaced with the word ‘revealed’ in the text.

Reviewer 2 Report

In this manuscript, the authors aim to highlight the contribution of neutrophil extracellular traps and NLRP3 inflammasome in atherosclerosis, inflammation and atherothrombosis. I have some minor comments as below:

1. More descriptions can be included for Figure 1.

2. More discussion should be provided on the dual interactions of neutrophil extracellular traps and NLRP3 inflammasome in atherosclerosis, inflammation and atherothrombosis.

3. Language use is very casual, professional language editing during revision is highly recommended.

Author Response

We are very grateful to the reviewers for pointing out very important and informative issues for the betterment of this manuscript. Please see point wise response in bold face.

Reviewer 2

 In this manuscript, the authors aim to highlight the contribution of neutrophil extracellular traps and NLRP3 inflammasome in atherosclerosis, inflammation and atherothrombosis. I have some minor comments as below:

  1. More descriptions can be included for Figure 1.

Sir, we have added more description for figure 1 in the text.

  1. More discussion should be provided on the dual interactions of neutrophil extracellular traps and NLRP3 inflammasome in atherosclerosis, inflammation and atherothrombosis.

We have added more description for dual interactions of NETs and NLRP3 inflammasome in atherosclerosis, inflammation and atherothrombosis as suggested.

  1. Language use is very casual, professional language editing during revision is highly recommended.

Thank you for suggesting the language editing. We asked a native English speaking colleague of Dr. Sarabjit (Co-author) at university of Loughborough, UK to review and improve language. We have incorporated these minor suggestions in our revised submission.